# MULTI-LAYERED 3D GARMENTS ANIMATION

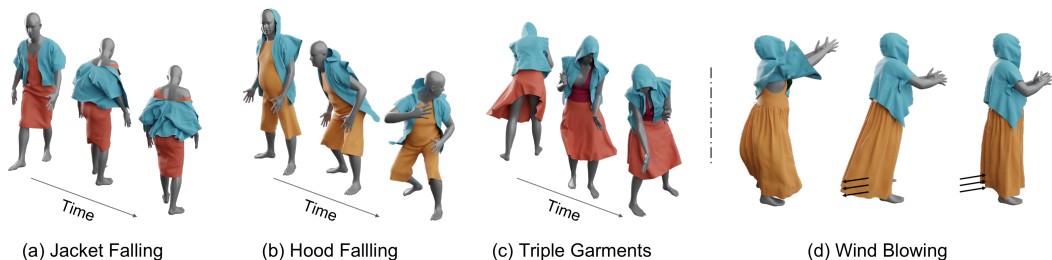

(a) Jacket Falling      (b) Hood Fallling      (c) Triple Garments      (d) Wind Blowing

Figure 1: We propose a new large-scale 3D garment animation dataset LAYERS, which improves over previous datasets by considering multi-layered 3D garments and more driving factors for garment animation, *e.g.*, environmental wind, besides human body movements. In (a)-(d) we show the new and realistic challenges covered in LAYERS but are omitted in previous datasets.

## ABSTRACT

Most existing 3D garment animation datasets are restricted to human bodies with single-layered garments. Even though cases with upper shirts and lower pants are included, only a few overlap areas among such garment combinations exist. Moreover, they often regard human body movement as the only driving factor that causes garment animation. Approaches developed on top of these datasets thus tend to model garments as functions of human body parameters such as body shape and pose. While such treatment leads to promising performance on existing datasets, it leaves a gap between experimental environments and real scenarios, where a body can wear multiple layered garments and the corresponding garment dynamics can be affected by environmental factors and garment attributes. Consequently, existing approaches often struggle to generalize to multi-layered garments and realistic scenarios. To facilitate the advance of 3D garment animation toward handling more challenging cases, this paper presents a new large-scale synthetic dataset called LAYERS, covering 4,900 different combinations of multi-layered garments with 700k frames in total. The animation of these multi-layered garments follows the laws of physics and is affected by not only human body movements but also random environmental wind and garment attributes. To demonstrate the quality of LAYERS, we further propose a novel method, LayersNet, for 3D garment animation, which represents garments as unions of particles and subsequently adopts a neural network to animate garments via particle-based simulation. In this way, the interactions between different parts of one garment, different garments on the same body, and garments against various driving factors, can be naturally and uniformly handled via the interactions of particles. Through comprehensive experiments, LayersNet demonstrates superior performance in terms of animation accuracy and generality over baselines. The proposed dataset, LAYERS, as well as the proposed method, LayersNet, will be publicly available.

## 1 INTRODUCTION

3D garment animation has been an active and important topic in computer graphics and machine learning, due to its great potential in various downstream tasks, including virtual reality, virtual try-on, gaming and film production. While this topic has been extensively studied in the past, generating realistic and faithful animation remains an open research question. In particular, existing approaches are still limited in modeling diverse garments of different topologies and appearances. In addition, the complex interactions between the garment and the human body under the challenging setting of multi-layer garments and with external environmental factors remain much less explored in the literature.

To support the development of data-driven approaches for 3D garment animation (Patel et al., 2020; Bertiche et al., 2020; Santesteban et al., 2021), researchers have built various datasets on real-life scans and synthetic data generated by Physically Based Simulation (PBS) (Narain et al., 2012; Li et al., 2018). However, most of existing datasets (Bertiche et al., 2020; Patel et al., 2020; Tiwari et al., 2020) consider only human bodies with single-layered garments, where each human body wears either a single dress or an upper t-shirt with lower pants that have limited overlap. The animation of multi-layered garments, such as a t-shirt with a jacket, that obey sophisticated physical dynamics, remain unexplored. In addition, in existing datasets, the moving human body is commonly regarded as the default and only driving factor in animating garments. Other factors, such as wind and friction, are left unconsidered. Such a simplification thus leads to a significant gap between experimental environments and real-world applications, making most approaches developed on top of these datasets less applicable in real life.

To bridge the gap between experimental environments and real-world applications and facilitate the advance of 3D garment animation, this paper introduces a new challenging dataset called LAYERS, muLti-lAYerEd gaRmentS dataset, which is carefully generated based on a simulation engine. LAYERS focuses on the animation of multi-layered garments, while also taking the wind, another important driving factor besides the human body, into consideration. Specifically, in LAYERS, multi-layered garments are prepared as combinations of inner and outer clothes, as shown in Figure 1. The inner and outer garments adopt different attribute values, e.g., bend stiffness and frictions. All garments on the same human body will interact with each other, constrained by the laws of physics. They are also simultaneously affected by the wind with randomly sampled direction and strength.

To demonstrate the quality of LAYERS, we further propose a novel data-driven method, dubbed as LayersNet, for multi-layered 3D garment animation. The core of LayersNet is a neural network based simulation system (Shao et al., 2022) that represents garments as unions of particles. Consequently, all kinds of interactions during garment animation, including the interactions between different parts of one garment, the interactions between different garments on the same body, and the interactions between garments and various driving factors, can be naturally and uniformly regarded as the interactions between particles. Hence, instead of being restricted to a specific driving factor (e.g., the human body) as previous methods, the proposed LayersNet possesses a strong generalization ability across diverse types of human body movements, multi-layered 3D garments, as well as driving factors. As the number of particles in LayersNet are considerably large when fine-grained details of garments and human bodies are preserved, we further exploit the redundancy of garments and extend LayersNet to establish a two-level structural hierarchy for garments where garments are made of patches, and patches are constituted of particles of a fixed configuration. Since the number of patches is much smaller than the number of particles, the interactions between all particles can be efficiently captured by the interactions of patches.

Our contributions can be summarized as follows:

1. We propose LAYERS, a large-scale and new dynamic dataset for 3D garment animation. The dataset focuses on multi-layered 3D garments, introducing random wind and friction as additional driving factors besides human body movements.

2. On top of LAYERS, we further propose LayersNet, a novel method for 3D garment animation that uniformly captures interactions among garment parts, different garments, as well as garments against driving factors. The notion of unifying various interactions as particle-based simulations is novel in the literature.

## 2 RELATED WORK

**3D Garment Datasets.** Publicly available 3D garment datasets are in great need. Existing datasets are generated either from synthesis (Pumarola et al., 2019; Patel et al., 2020; Santesteban et al., 2021; Bertiche et al., 2020) or real-world scans (Zhang et al., 2017; Zheng et al., 2019; Ma et al., 2020; Tiwari et al., 2020; Cai et al., 2022). For synthetic datasets, 3DPeople (Pumarola et al., 2019) contains multi-view images including RGB, depth, normal, and scene flow data. TailorNet (Patel et al., 2020) provides a synthetic dataset with 20 different garments simulated in 1,782 static SMPL poses for nine body shapes. Santesteban et al. (2021) contributes a dataset composed of two different garments simulated on 56 human motion sequences with 17 body shapes. Cloth3D (Bertiche et al.,

2020) is the largest synthetic dataset, with 11,300 outfits generated from the combinations of several prototypes, such as t-shirts, tops, trousers and skirts. Most existing datasets only contain single-layered 3D garment models. Even though there exist combinations with multiple garments, such as an upper t-shirt and lower pants in Multi-Garment Net Bhatnagar et al. (2019), there are very few overlapping areas among different cloth pieces. Recently, Layered-Garment Net Aggarwal et al. (2022) proposes a static multi-layered garments dataset in 7 static poses for 142 bodies to generate layers of outfits from single image. However, the garments in Layered-Garment Net, which are mostly skinning clothes, do not follow physics laws and the interpenetration is solved by simply forcing penetrated vertices out of inner garments.

To our best knowledge, LAYERS is the first dataset containing dynamic multi-layered 3D garments, e.g., the human model wears a dress and outer jacket which deform according to the body motions. Different layers of garments have different attributes and interact with each other, obeying the laws of physics. Moreover, we introduce wind as extra driving factor to animate the garments, enriching their dynamics given similar human movements. Our dataset includes all necessary 3D information, which is able to easily generalize to other tasks, such as reconstructions from single images.

**Data-driven Cloth Model.** Most existing approaches aim to estimate a function that outputs the deformations of garments for any input. A common strategy is to learn a parametric garment model to deform the corresponding mesh templates. For example, garments are modeled as functions of human pose (Wang et al., 2019), shape (Vidaurre et al., 2020), pose-and-shape (Bertiche et al., 2020; 2021; Tiwari & Bhowmick, 2021), motions (Santesteban et al., 2021), garment type (Ma et al., 2020; Patel et al., 2020). The approaches mentioned above rely heavily on SMPL-based human models and animate garments by the blend weight according to the registered templates. The generalization of such approaches is limited to skinning clothes. Some recent studies explore bone-driven motion networks (Pan et al., 2022) to animate loose garments by virtual bones, which can be regarded as extra anchors besides SMPL model. SCALE (Ma et al., 2021) adopts local elements to model registered garments based on minimum-clothed human. To handle obstacles with arbitrary topologies, N-Cloth (Li et al., 2022) predicts garments deformations given the states of initial garments and target obstacles. Other studies Shen et al. (2020); Zhang et al. (2022) generate 3D garments based on UV maps. SimulCap Yu et al. (2019) segments garments into upper and lower clothes as multiple separated meshes. SMPLicit Corona et al. (2021) generates garments by controlling the clothes' shapes and styles, but intersection-free reconstruction is not guaranteed.

In contrast, our data-driven method LayersNet animates garments by inferring garments' future positions through the interactions between garment particles and other driving factors. Since the driving factors are also represented by particles, the garment animation thus equals to simulate particle-wise interactions, which is shape-independent and is highly generalizable to unseen scenarios.

**Physics Simulation by Neural Network.** Learning-based methods for physics simulation can be applied to different kinds of representations, e.g., approaches for grid representation (Thuerey et al., 2020; Wang et al., 2020), meshes (Nash et al., 2020; Qiao et al., 2020; Weng et al., 2021; Pfaff et al., 2021), and particles (Li et al., 2019; Ummenhofer et al., 2020; Sanchez-Gonzalez et al., 2020; Shao et al., 2022). Some methods include Graph Neural Network (GNN)-based methods (Li et al., 2019; Sanchez-Gonzalez et al., 2020; Pfaff et al., 2021). Transformer-based methods (Shao et al., 2022) adopt modified attention to recover interactions' semantics. Other work Liang et al. (2019) designs algorithms to accelerate gradient computation for collision response as plug-ins for neural network.

Our LayersNet follows TIE in the notion of modeling particle-wise interactions, which is topology-independent and easy to generalize to unseen scenarios. In contrast to TIE, we exploit the redundancy of garments and establish a two-level hierarchy structure for them, where garments are made of deformable patches. Our method also differs in learning to predict the patches' dynamics by interacting with neighbor patches and other driving factors. We devise a decoder to learn a topology-independent descriptor for each patch, enhancing the generalization abilities to unseen scenarios.

## 3  LAYERS DATASET

Most existing datasets are limited to single-layered garments driven only by human bodies. Different garments, such as the upper T-shirt and lower pants, rarely interact with each other. Consequently,

Table 1: We compare LAYERS with existing 3D datasets. Our dataset is composed of multi-layered clothes, with unique attribute data, such as stiffness and friction, attached to each garment. Moreover, we include data of wind with its strength and direction randomly sampled. *[1]: 3DPeople (Pumarola et al., 2019) does not specify the exact number of garments, while it claims to dress each subject with different outfits. *[2]: The multi-layered garments in Layered-Garment do not follow physics laws and the penetrated vertices are forced to move out of inner garments in hard-coded manner.

| Dataset | Dynamics | Subjects | Garments | Multi-layered | Attributes | Wind |
|---|---|---|---|---|---|---|
| 3DPeople (Pumarola et al., 2019) | | 80 | *[1] | | N/A | |
| TailorNet (Patel et al., 2020) | | 9 | 20 | | N/A | |
| Cloth3D (Bertiche et al., 2020) | ✓ | 8.5K | 11.3K | | 4 | |
| Layered-Garment (Aggarwal et al., 2022) | | 142 | 101 | *[2] | N/A | |
| LAYERS (Ours) | ✓ | 4.9K | 9.9K | ✓ | 9.9K | ✓ |

the problem can be easily solved by modeling garments as functions of human bodies and considering only single-layered outfits predictions (Patel et al., 2020; Bertiche et al., 2021).

Generating a dataset with multi-layered garments is non-trivial – interpenetration between garments should be avoided, and their dynamics should obey the physics rules. Thanks to recent developments in physics-based methods, several software, such as Blender[1], can infer the interactions among different clothes and generate faithful garments with multiple layers.

The proposed LAYERS is built with Blender. It is the first dynamic multi-layered garments dataset that considers the wind factor beyond just human bodies. To construct the dataset, we first collect the garment templates from SewPattern(Korosteleva & Lee, 2021), which includes various types of garments, such as jackets with hood, and dresses with waist belts. Then, we generate multi-layered combinations with outer-layer and inner-layer clothes. Each combination of multi-layered garments is then draped to SMPL human body (Loper et al., 2015). This is followed by a warm-up simulation in Blender to resolve interpenetrations. Finally, we simulate the dynamics of garments given the human motion sequences (Mahmood et al., 2019) and sampled winds. Specifically, after we drape the garments to a human body model, we scale up the human mesh and garment mesh ten times the real-world size before simulation. This strategy can reserve more high-frequency details in Blender. We compare our dataset with existing datasets in Table 1.

Since our dataset includes the 3D meshes and attributes of garments, as well as the detailed scene settings for each sequence, LAYERS can be easily extended to other formats of data to facilitate the explorations of alternative topics, such as optical flow estimations, 3D reconstructions from images, and physics parameters estimations. In the following, we detail the key settings in LAYERS.

**Multi-layered Garments.** Each multi-layered outfit is composed of inner and outer outfit. In LAYERS, the outer outfit is either a jacket or a jacket with a hood, giving us a clear view of interactions from inside and outside. Inner outfits refer to whole-body outfits, such as dresses, jumpsuits, and t-shirts with pants or skirts. We generate 4,900 combinations of multi-layered garments, with 9,872 different garments in total. The garment templates are in high fidelity, with vertices ranging from 5,000 to more than 15,000 for each garment, enabling us to capture more details in simulation.

The main challenge is to have interpenetration-free simulations for multiple objects. To achieve that, we first drape the multi-layered outfit to SMPL human body in T-pose, followed by a warm-up simulation to solve the interpenetrations among garments. We adopt a large collision distance to ensure all the interpenetrations are resolved. Afterwards, we merge the garments into one garment mesh and conduct a simulation driven by the human body and wind. Since all garments belong to one mesh after merging, the interactions among garments are computed through the self-collision mechanism in Blender, which generates interpenetration-free results in simulation.

**Wind.** Most existing datasets simplify real-life scenarios through driving the animation of garments only by human bodies. To enrich the settings and enable researchers to further explore garment animations driven by multiple factors, in LAYERS we introduce randomly sampled wind, a common and obvious force field to influence the animation. Specifically, we randomly select several spans of frames in a sequence, and apply winds of different directions and strengths as force fields. The directions and strengths are uniformly sampled (0 to 400 in Blender). Within each span,

---

[1]https://www.blender.org/

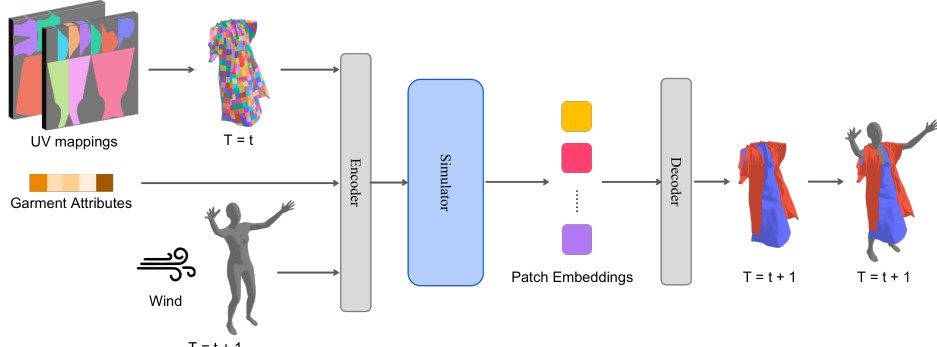

Figure 2: Overview of LayersNet. Given driving factors at time $t + 1$, i.e., the human body model and environmental wind in our study, LayersNet aims to animate target garments at time $t$ and predicts the new states of garments at time $t + 1$. Human body model, wind, garments and corresponding attributes are represented by particles, where we adopt abstract particles to denote the wind and garments' attributes. We further establish a two-level structural hierarchy for garments, as shown on the top left of the figure, where garments are made of patches, and patches are composed of particles of a fixed configuration given the UV mappings. Then we encode the particles and model the interactions among them by a simulator, which outputs the embeddings for each patch. Finally, we apply a decoder to decode the patches into corresponding vertices at time $t + 1$.

we make a reasonable assumption that the wind would affect the whole 3D space, where wind's direction and strength remain constant.

**Garments' Attributes.** Existing datasets cover a limited choice for garment attributes, e.g., cotton or fabric. Tasks like physics parameter estimations could barely benefit from those datasets. To make garments animations more diverse and flexible, we uniformly sample different garments' attributes, such as mass, stiffness, and friction. In addition, we introduce different attributes to the inner and outer outfits, leading to more varieties.

**Human Motion Sequences.** We adopt the SMPL-based human motion sequences from CMU MoCap in AMASS (Mahmood et al., 2019), which includes 2,600 sequences with 30FPS in total. During simulation, we randomly sample the human shapes and genders for each sequence and extract sub-sequences with a maximum of 600 frames. To accurately simulate garments on human body, collision-free human meshes are required to avoid invalid simulations. Since our dataset focuses on garments generations, we adopt linear regressions to solve the self-collisions from SMPL models (Loper et al., 2015) and leave a minimum gap of 0.004 meters before scaling up the human mesh. We skip the unresolvable collisions and discard the corresponding frames.

## 4 METHODOLOGY

Our goal is to faithfully animate the garments regardless of the garments' topology and the type of driving factors, where the latter include the rigid human bodies and winds in our case. To this end, we propose LayersNet to animate garments in a simulation manner. The novelty of LayersNet is that we view garment animations driven by human bodies and winds as interactions between particles. This unified perspective allows our framework to exploit the semantics of interactions among all particles, e.g., the energy transition when constrained by physics laws. In addition, the animation of garments becomes shape-independent and highly generalizable. Figure 2 shows an overview of LayersNet. In the following, we first formulate garment animation in the form of particle simulation, followed by an explanation of our patch-based garment model, and the introduction of the simulation pipeline to animate garments.

### 4.1 LAYERSNET

**Problem Formulation.** We denote each mesh at time $t$ by $M^t = \{\boldsymbol{V}^t, \boldsymbol{E}^M, \boldsymbol{E}^W\}$, where $\boldsymbol{V}^t = \{\boldsymbol{x}_i^t, \dot{\boldsymbol{x}}_i^t, \ddot{\boldsymbol{x}}_i^t\}_{i=1}^N$ are the vertices' positions, velocities, and accelerations, and $\boldsymbol{E}^M$ denote the mesh edges. $\boldsymbol{E}^W$ are the world space edges (Pfaff et al., 2021), where we dynamically connect node $i$ and node $j$ if $|\boldsymbol{x}_i^t - \boldsymbol{x}_j^t| < R$, excluding node pairs already exist in the mesh. In a particle-based system, each mesh is represented by particles, which are the corresponding vertices from the mesh. During simulation, particle $i$ and particle $j$ will interact with each other iff an edge $e_{ij} \in \boldsymbol{E}^M \cup \boldsymbol{E}^W$

connects them. The interactions guided by $\boldsymbol{E}^M$ enable learning internal dynamics of mesh, while interactions indicated by $\boldsymbol{E}^W$ serve to compute external dynamics such as collisions.

We adopt abstract particles to represent the garments' attributes and the wind. In particular, we use $\boldsymbol{a}_g$ to denote each garment's attribute, such as the friction and stiffness, and $\boldsymbol{w}^t$ to denote the wind. Since the wind has constant strength in the whole 3D space, we use the quaternion rotation $\boldsymbol{q}^t$ and the strength $s^t$ to represent the wind as $\boldsymbol{w}^t = \{\boldsymbol{q}^t, s^t\}$. In this way, given the human body and wind at $t+1$ as well as their previous $h$ states, we aim to predict the garments' states at time $t+1$ given the current states at $t$ and corresponding previous meshes $\{M^{t-1}, \cdots, M^{t-h}\}$. In practice, we choose $h = 1$ in all experiments. Our approach can be described as:

$$\hat{\boldsymbol{V}}_g^{t+1} \quad = \quad \phi(\boldsymbol{a}_g, \{M_g^{t-i}, M_b^{t+1-i}, \boldsymbol{w}^{t+1-i}\}_{i=0}^h), \tag{1}$$

where $M_g^t$ and $M_b^{t+1}$ are the meshes of garments and human body, respectively, $\phi(\cdot)$ is the simulator and runs recursively during predictions, and $\hat{\boldsymbol{V}}_g^{t+1}$ is the garment's new vertices' states at time $t+1$.

**Patch-based Garment Model.**    Inspired by Ma et al. (2021), we establish a two-level structural hierarchy for garments and represent each garment by patches. Patch modeling holds several advantages. First, as basic units to represent garments, patches are topology independent. By modeling the dynamics of each patch, our model is more flexible and generalizable to unseen garments. Second, instead of simulating each vertex in a mesh, simulating patches signficantly reduces the computational overhead, especially when the mesh is of high-fidelity.

Formally, we find a mapping $q(\cdot)$ to map the vertex-based mesh to patch-based representation by:

$$P_g^t \quad = \quad q(M_g^t), \tag{2}$$

where $P_g^t = \{\boldsymbol{V}_p^t, \boldsymbol{E}_p^M, \boldsymbol{E}_p^W\}$. The patches' states $\boldsymbol{V}_p^t$ are the averaged vertices' states within the patches, and $\boldsymbol{E}_p^W$ are computed given $\boldsymbol{V}_p^t$. The mapping $q(\cdot)$ is based on the garments' uv maps as shown in Figure 2. In this way, our method can be updated as:

$$\hat{\boldsymbol{V}}_g^{t+1} \quad = \quad \phi(\boldsymbol{a}_g, \{P_g^{t-i}, M_b^{t+1-i}, \boldsymbol{w}^{t+1-i}\}_{i=0}^h), \tag{3}$$

**Simulation-based Garment Animation.**    After collecting the set of particles, including the patch particles of garments, the vertex particles of human body, as well as the abstract particles of garment attributes and environmental wind, we can then animate garments by predicting the future state of each particles through particle-based simulation, which is topology independent and highly generalizable. While our method is orthogonal to the choice of particle simulator, in practice we adopt TIE (Shao et al., 2022) as our simulator due to its promising results and high computational efficiency. Specifically, it assigns each particle $i$ three tokens, namely a state token $\boldsymbol{v}_i$, a sender token $\boldsymbol{s}_i$ and a receiver token $\boldsymbol{r}_i$. The sender token $\boldsymbol{s}_i$ describes how the particle $i$ influence others, and the receiver token $\boldsymbol{r}_i$ indicates how the particle $i$ can be affected. The updating formulas of all tokens can be summarized as follow:

$$\boldsymbol{s}_i \quad = \quad W_s \boldsymbol{v}_i, \qquad \boldsymbol{r}_i = W_r \boldsymbol{v}_i, \tag{4}$$

$$\boldsymbol{s}_j' \quad = \quad \frac{\boldsymbol{s}_j - \mu_{\boldsymbol{s}_j}}{\sigma_{\boldsymbol{r}_i \boldsymbol{s}_j}}, \qquad \boldsymbol{r}_i' = \frac{\boldsymbol{r}_i - \mu_{\boldsymbol{r}_i}}{\sigma_{\boldsymbol{r}_i \boldsymbol{s}_j}}, \tag{5}$$

$$\boldsymbol{f}_{\boldsymbol{r}_i, \boldsymbol{s}_j} \quad = \quad \boldsymbol{r}_i' + \boldsymbol{s}_j', \tag{6}$$

$$\boldsymbol{v}_i' \quad = \quad \sum_j \omega_{ij} \boldsymbol{f}_{\boldsymbol{r}_i, \boldsymbol{s}_j}, \tag{7}$$

$$\omega_{ij} \quad = \quad \text{softmax}((W_Q \boldsymbol{v}_i)^\top \boldsymbol{f}_{\boldsymbol{r}_i, \boldsymbol{s}_j}), \tag{8}$$

where $\mu_{\boldsymbol{r}_i}, \mu_{\boldsymbol{s}_j}$ are the means of tokens $\boldsymbol{r}_i, \boldsymbol{s}_j$ respectively, and $\sigma_{\boldsymbol{r}_i \boldsymbol{s}_j}$ is the standard deviation. In practice, we generate two different attention masks indicated by $\boldsymbol{E}^M$ and $\boldsymbol{E}^W$ for different heads in multi-head attention.

Formally, the simulator can be described as

$$\boldsymbol{E}_p^{t+1} \quad = \quad f(\boldsymbol{a}_g, \{P_g^{t-i}, M_b^{t+1-i}, \boldsymbol{w}^{t+1-i}\}_{i=0}^h). \tag{9}$$

To recover each vertex's details within the corresponding patch, we apply a decoder as:

$$\hat{\boldsymbol{a}}_i^{t+1} \quad = \quad g([\boldsymbol{v}_i^t, \boldsymbol{e}_{p,i}^{t+1}, \boldsymbol{v}_{h,i}^{t+1}]), \tag{10}$$

$$\hat{\boldsymbol{v}}_i^{t+1} \quad = \quad \Delta t \cdot \hat{\boldsymbol{a}}_i^{t+1} + \boldsymbol{v}_i^t, \qquad \hat{\boldsymbol{p}}_i^{t+1} = \Delta t \cdot \hat{\boldsymbol{v}}_i^{t+1} + \boldsymbol{p}_i^t, \tag{11}$$

where we concatenate $i$-th vertex's state $\boldsymbol{v}_i^t$ with its corresponding patch embedding $\boldsymbol{e}_{p,i}^{t+1}$ and the states of the nearest point on human mesh $\boldsymbol{v}_{h,i}^{t+1}$ as inputs, and output the new acceleration $\hat{\boldsymbol{a}}_i^{t+1}$ at time $t+1$. We then calculate the corresponding position and velocity at time $t+1$ given the time interval $\Delta t$ between each frame.

## 4.2 TRAINING DETAILS

To train a simulation model, we first apply a standard mean square error (MSE) loss on the positions of vertices as:

$$\mathcal{L}_m^{t+1} \quad = \quad \frac{1}{N} \sum_i \|\hat{\boldsymbol{p}}_i^{t+1} - \boldsymbol{p}_i^{t+1}\|_2^2, \tag{12}$$

where $\boldsymbol{p}_i^{t+1}$ is the ground truth at time $t+1$ and $N$ is the number of vertices. We adopt a loss term for the vertex normal to maintain the smoothness and consistence of the garments:

$$\mathcal{L}_n^{t+1} \quad = \quad \frac{1}{N} \sum_i \|\hat{\boldsymbol{n}}_i^{t+1} - \boldsymbol{n}_i^{t+1}\|_2^2, \tag{13}$$

where $\hat{\boldsymbol{n}}_i^{t+1}$ and $\boldsymbol{n}_i^{t+1}$ are the vertex normal for prediction and ground truth respectively. To further reduce the collision rates between garments and human bodies, we adopt a collision loss:

$$\mathcal{L}_c^{t+1} \quad = \quad \frac{1}{N_c} \sum_i \left( d_\epsilon - \min \left( (\hat{\boldsymbol{p}}_i^{t+1} - \boldsymbol{p}_h^{t+1}) \cdot \boldsymbol{n}_h^{t+1}, d_\epsilon \right) \right)^2, \tag{14}$$

where $\boldsymbol{p}_h^{t+1}$ is the nearest point to $\hat{\boldsymbol{p}}_i^{t+1}$ on the human mesh, $\boldsymbol{n}_h^{t+1}$ is the normal vector of point $\boldsymbol{p}_h^{t+1}$, $N_c$ is the number of collided vertices, and $d_\epsilon$ is the minimum distance of penetration. Thus, for predictions at time $t+1$, our training loss is written as:

$$\mathcal{L}^{t+1} \quad = \quad \lambda_m \mathcal{L}_m^{t+1} + \lambda_n \mathcal{L}_n^{t+1} + \lambda_c \mathcal{L}_c^{t+1}, \tag{15}$$

During training, we predict the garments' positions for two future timestamps, namely $t+1$ and $t+2$. The final loss $\mathcal{L}$ is

$$\mathcal{L} \quad = \quad \mathcal{L}^{t+1} + \mathcal{L}^{t+2}. \tag{16}$$

## 5 EXPERIMENTS

### 5.1 BASELINE AND IMPLEMENTATION DETAILS

We implement DeePSD (Bertiche et al., 2021) and MGNet (Zhang et al., 2022) as our baselines. DeePSD achieves state-of-the-art performance in terms of 3D garment animations. Importantly, it claims to support the animations of multi-layered garments. We mainly compare DeePSD with our model and make the following extensions to DeePSD: 1. we add wind as extra inputs; 2. we add the collision loss between different layers of garments for multi-layered clothes settings. Since MGNet is a garment-specific model for single-layered clothes, we compare MGNet with only inner garments. We also include the wind as extra feature map. All models are trained with ten epochs. We do not apply any postprocessing for both training and predicting. During evaluation, we calculate errors as the mean of Euclidean errors for each frame, then average the errors of all frames within each sequence. The final results are the mean of errors from all sequences.

### 5.2 ABLATION STUDY ON LAYERS

To investigate the influence of multi-layered garments and random wind in LAYERS, we divide our dataset into four different splits as shown in Table 2 : tight inner garments without wind (T);

Table 2: To analyze the challenges in LAYERS , we sample four splits from our dataset: inner garments are tight clothes without wind (T); inner garments are tight clothes with strong wind (T+W); inner garments are loose clothes without wind (L); inner garments are loose clothes with strong wind (L+W). Specifically, jackets are either with or without hood, while dress are either with or without waist belt. Notice that we group winds with a strength less than 50 as not windy, where the wind has little influence on the garments.

| Components | Tight (T) | Tight+Wind (T+W) | Loose (L) | Loose+Wind (L+W) |
|---|---|---|---|---|
| Inner Garments | Jumpsuit | Jumpsuit | Dress | Dress |
| Outer Garments | Jacket | Jacket | Jacket | Jacket |
| Wind Strength | $\leq 50$ | $> 250$ | $\leq 50$ | $> 250$ |

Table 3: Euclidean errors (mm) on four splits. To display the challenges brought by the outer garments and the interactions between layers of clothes, we further train models with only the inner garments as marked by * in the table. MGNet has worse generalization abilities due to garment-specific design. LayersNet has slightly higher errors since the inner garments simulated together with outer clothes do not follow physics laws by themselves. LayersNet achieves superior and robust performance on all splits with multi-layered garments.

| Methods | Tight (T) | Tight + Wind (T+W) | Loose (L) | Loose + Wind (L+W) |
|---|---|---|---|---|
| DeePSD* | **225.3±106.4** | **239.5±103.9** | 501.3±300.1 | 577.5±373.9 |
| MGNet* | 5219.2±1565.8 | 5186.8±1754.8 | 4432.7±1438.0 | 4595.0±1215.2 |
| LayersNet* | 260.3±254.4 | 278.6±328.6 | **378.0±293.0** | **363.6±311.4** |
| DeePSD | 1068.2±693.8 | 2782.6±1239.7 | 868.0±495.6 | 1707.9±503.4 |
| LayersNet(Ours) | **611.3±544.3** | **578.8±576.2** | **603.0±529.1** | **572.5±469.5** |

tight inner garments with strong wind (T+W); loose inner garments without wind (L); loose inner garments with strong wind (L+W). Each split contains 36K frames for training, 2K frames for validation, and 2K frames for test. Note that the strength of the wind ranges from 0 to 400. We group winds with a strength less than 50 as not windy, where the wind has little influence on the garments. We consider winds with a strength more than 250 as strong wind. Since the outer garments exhibit more flexible dynamics, such as falling off or waving in the air, animating them is already a challenging task for existing methods, let alone considering the interactions with inner garments. To further simplify our dataset and have a better comparison with existing synthetic dataset, Cloth3D (Bertiche et al., 2020), we first exclude the outer garments on all splits and train models with only inner garments, which is indicated in the first three rows of Table 3.

When trained with only inner garments, DeePSD achieves reasonable performance comparing with that when it is trained on Cloth3D (Bertiche et al., 2020), suggesting that the settings of Cloth3D are similar to our simplified data settings. MGNet fails in our dataset due to the garment-specific design and low generalization abilities. LayersNet has lower errors especially on split L and L+W, suggesting the effectiveness and higher generalization abilities of animating loose clothes, Please refer to Appendix for more details and qualitative comparisons. On split T+W and L+W, DeePSD shows higher errors due to the random wind. Since jumpsuits in splits T and T+W are tight garments, the wind has less influence on them.

When trained with multi-layered garments, the Euclidean errors by DeePSD increase dramatically, especially with the influence of wind in split T+W and L+W, suggesting the challenges brought by multi-layered garments. The high errors on split T+W and L+W, compared with split T and L, respectively, suggest that DeePSD is less generalizable to driving factors beyond human bodies. In contrast, LayersNet achieves superior performance on all splits with both inner and outer garments. The Euclidean errors are close to each other on different splits, suggesting that our model is more robust to the garments' various topologies as well as the driving factors beyond human bodies.

## 5.3 GARMENT ANIMATION

We sample 50K frames from LAYERS for training, 6K frames for validation, and 6K frames for test. There is no overlapping among different sets of samples. All the samples are composed of both inner and outer garments, as well as random wind as the external factor.

The vanilla DeePSD without collision loss exhibits high Euclidean errors on all types of garments. When adding collision loss including collisions between layers of garments, the collision rates are reduced. Nonetheless, DeePSD does not improve in terms of Euclidean errors. Since the vanilla DeePSD has difficulties in learning reasonable dynamics of garments, the collision terms introduce more noise while training DeePSD , pushing the garments away from the bodies. In contrast, LayersNet achieves superior performance in terms of Euclidean errors, suggesting the effectiveness of our simulation-based methods. Moreover, our LayersNet  is more generalizable and shows more

Table 4: Euclidean error (mm) on sampled LAYERS with maximum sequence length of 35 frames. The collision rates between different layers of garments are shown under **L-Collision**, while the collision rates between garments and human bodies are shown under **H-Collision**. Models trained with collision loss are marked by +. When training DeePSD with collision loss, we extend Equation 14 and include the collisions between different layers of garments as extra loss term. In contrast, our model only applies the basic collision loss between garments and human bodies following Equation 14. Our LayersNet achieves superior results on all types of garments. Even without explicitly punishing collisions between layers of garments, LayersNet still achieves low collision rates among garments and makes good balance between Euclidean errors and penetrations.

| Methods | Jacket | Jacket + Hood | Dress | Jumpsuit | Skirt |
|---|---|---|---|---|---|
| DeePSD | 2863.0±881.2 | 2956.7±799.8 | 2606.3±792.0 | 2876.2±708.3 | 2498.6±618.6 |
| DeePSD+ | 3609.7±1147.0 | 3434.4±781.9 | 4046.6±962.9 | 4773.1±1120.9 | 4232.6±1713.9 |
| LayersNet(Ours) | 717.6±609.8 | 577.5±458.1 | **448.2±452.2** | **277.3±293.1** | **274.6±94.4** |
| LayersNet+(Ours) | **684.9±554.9** | **566.2±425.4** | 501.2±466.9 | 321.1±274.7 | 378.6±143.0 |

| Methods | Pants | T-shirt | Overall | L-Collision | H-Collision |
|---|---|---|---|---|---|
| DeePSD | 3075.1±117.7 | 2618.5±729.1 | 2851.8±696.4 | 23.82%±11.25% | 18.36%±6.74% |
| DeePSD+ | 5260.4±1283.4 | 4145.9±994.3 | 3907.6±790.6 | **3.82%±3.72%** | **0.63%±0.83%** |
| LayersNet(Ours) | **291.2±301.7** | **272.4±198.2** | 560.6±452.2 | 4.51%±2.98% | 10.01%±5.62% |
| LayersNet+(Ours) | 349.3±238.4 | 331.7±226.9 | 567.2±432.8 | 4.94%±2.67% | 3.58%±2.83% |

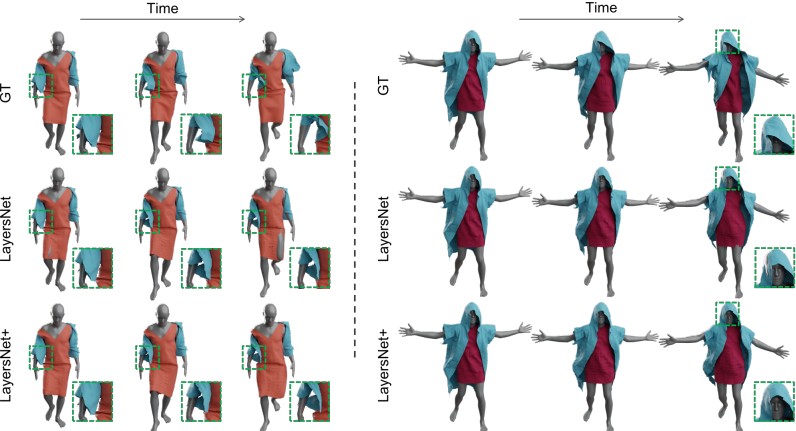

Figure 3: Qualitative results by LayersNet. The left sequence shows a human model walking down some steps. The magnified regions highlight the vivid dynamics of the jacket. Human model in the right sequence is moving towards her left. Without collision loss, LayersNet generates some body-to-cloth penetrations near the thigh and even head on the last frame of the right sequence. When trained with collision loss, LAYERS, which is marked by +, reduces the collision rates more obviously. Even though we do not explicitly penalize the collisions among different layers of garments, LayersNet is able to solve those collisions implicitly through exchanging semantics by interactions, suggesting the robustness and effectiveness of LayersNet.

robust performance across different types of garments. When adding collision loss as mentioned in Equation 14, the body-to-cloth collisions and human bodies are reduced. Though we never explicitly penalize collisions between different layers of garments, the collision rates among clothes are low. Since the key idea of simulation is to model the interactions among objects, such as the energy transition and collisions, LayersNet can resolve collisions implicitly. We show the qualitative results by our LayersNet in Figure 3. More qualitative comparisons can be found in Appendix. LayersNet with collision loss, which is marked by +, shows fewer body-to-garment penetrations, suggesting the effectiveness and robustness of our model.

## 6 CONCLUSION

We have presented a new large-scale synthetic dataset called LAYERS, which covers 4,900 different combinations of multi-layered garments with 700K frames in total. The animations of multi-layered garments follow the laws of physics, allowing the interactions among different layers of garments. In addition, LAYERS takes the environmental wind, another important driving factors besides human body, into consideration to animate garments. To demonstrate the quality of LAYERS, we further propose LayersNet, a simulation-based method for garment animations. We model the various driving factors as (abstract) particles, while represent garments as unions of particles as patches. The animations of garments driven by different factors are naturally and uniformly achieved via modeling the interactions among particles. As shown by the experiments, our model achieves superior and robust performance with compelling abilities in generalization.

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

# A APPENDIX

## A.1 LAYERS DATASET

Although Blender does not support collisions among multiple objects, it is able to solve collisions within one object. Thus, by merging multiple garments as a single mesh, we regard the collisions among different garments as the interactions within one mesh, which can be solved by Blender. In other words, different layers of garments will interact with each other following the physics rules.

Before simulation, we properly dress the human body in T-pose and scale up all objects 10 times the real-world size, which can preserve more details of the garments such as wrinkles.

We adopt different garment attributes to different layers of clothes. Specifically, we uniformly sample the following attributes: vertex mass from 0.2 to 0.8; stiffness of tension, compression, shear, and bending from 15 to 100; friction from 40 to 80.

**Quantitiave comparison with existing synthetic dataset.** We train our implementation of DeePSD on Cloth3D following the original paper (Bertiche et al., 2021). Results are shown in Table 5, suggesting that our implementation of DeePSD is similar to the official one. We sample four splits from LAYERS as mentioned in main text. Specifically, the inner garments in split T and T+W are jumpsuits, while those in split L and L+W are dress. The outer garments in all splits are either jackets or jackets with hood. For convenience, we copy the table in main text for reference as shown in Table 6 and Table 7. Table 7 shows the results trained on four splits of our dataset. Specifically, the results on the first three rows, where models are marked by *, are obtained by training models with only inner garments. This setting is most similar to Cloth3D's settings. The remaining results are obtained by training on both inner and outer garments on all splits. Notice that we scale up the human mesh and garment mesh 10 times the real-world size, the corresponding errors are also scaled up. Thus, DeePSD* achieves similar results on both datasets: the Euclidean errors of jumpsuit and dress are similar on both LAYERS and Cloth3D, suggesting that the quality of our dataset is not worse than Cloth3D. In addition, when comparing DeePSD* with DeePSD, DeePSD achieves higher errors, which mainly come from the outer garments. MGNet fails in LAYERS due to the garment-specific design and low generalization abilities as shown in Figure 4. Our LayersNet achieves reasonable results especially on loose garment settings, suggesting the effectiveness and higher generalization abilities of animating loose garments.

Table 5: We verify our implementation of DeePSD on Cloth3D accroding to official paper (Bertiche et al., 2021). The results are similar to original paper, suggesting that our implementation of DeePSD is similar to official one.

| Method | T-shirt | Top | Trousers | Skirt | Jumpsuit | Dress |
|---|---|---|---|---|---|---|
| DeePSD | 25.01±20.94 | 16.90±15.38 | 20.02±8.50 | 20.43±31.10 | 24.31±6.36 | 42.10±21.41 |

Table 6: To analyze the challenges in LAYERS , we sample four splits from our dataset: inner garments are tight clothes without wind (T); inner garments are tight clothes with strong wind (T+W); inner garments are loose clothes without wind (L); inner garments are loose clothes with strong wind (L+W). Specifically, jackets are either with or without hood, while dress are either with or without waist belt. Notice that we group winds with a strength less than 50 as not windy, where the wind has little influence on the garments.

| Components | Tight (T) | Tight+Wind (T+W) | Loose (L) | Loose+Wind (L+W) |
|---|---|---|---|---|
| Inner Garments | Jumpsuit | Jumpsuit | Dress | Dress |
| Outer Garments | Jacket | Jacket | Jacket | Jacket |
| Wind Strength | $\leq 50$ | $> 250$ | $\leq 50$ | $> 250$ |

Table 7: Euclidean errors (mm) on four splits. To display the challenges brought by the outer garments and the interactions between layers of clothes, we further train models with only the inner garments as marked by * in the table. MGNet has worse generalization abilities due to garment-specific design. LayersNet has slightly higher errors since the inner garments simulated together with outer clothes do not follow physics laws by themselves. LayersNet achieves superior and robust performance on all splits with both inner and outer garments.

| Methods | Tight (T) | Tight + Wind (T+W) | Loose (L) | Loose + Wind (L+W) |
|---|---|---|---|---|
| DeePSD* | **225.3±106.4** | **239.5±103.9** | 501.3±300.1 | 577.5±373.9 |
| MGNet* | 5219.2±1565.8 | 5186.8±1754.8 | 4432.7±1438.0 | 4595.0±1215.2 |
| LayersNet* | 260.3±254.4 | 278.6±328.6 | **378.0±293.0** | **363.6±311.4** |
| DeePSD | 1068.2±693.8 | 2782.6±1239.7 | 868.0±495.6 | 1707.9±503.4 |
| LayersNet(Ours) | **611.3±544.3** | **578.8±576.2** | **603.0±529.1** | **572.5±469.5** |

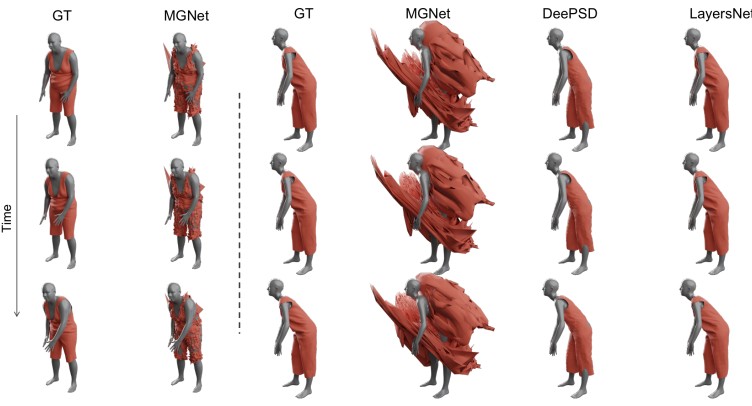

Figure 4: The left are the training samples while the right is test samples. All samples are from the split T, where we train models with only inner garments (jumpsuit). MGNet is able to generate 3D garments on training examples on the left while has difficulties to generalize to unseen examples in test set due to the garment-specific design. DeePSD has faithful predictions on simplified dataset. Our LayersNet faithfully rollouts 3D garments when only with inner clothes.

**Qualititative comparisons.** We include more qualitative comparisons in this section. As shown in Figure 4 when trained with only inner garments (jumpsuit), DeePSD (Bertiche et al., 2021) is

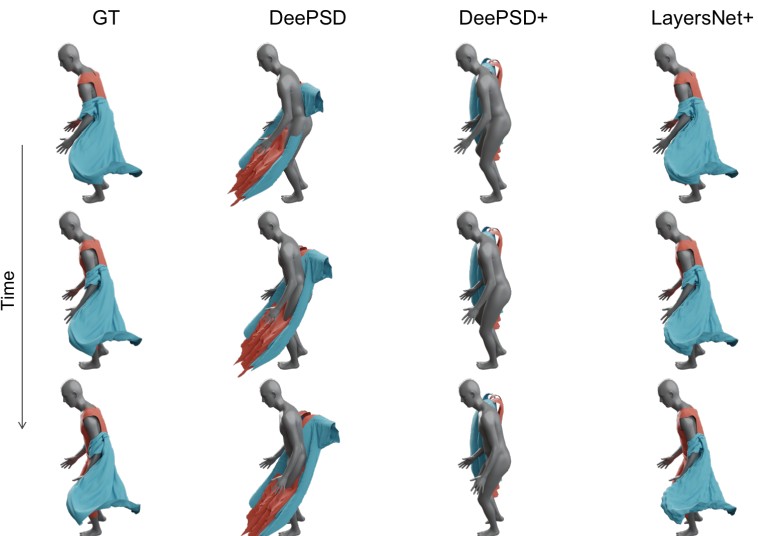

Figure 5: The outer garments are more flexible in our dataset and are able to respond to different garment attributes, such as friction. In this sample, the jacket falls off the shoulder due to the motion of human and small friction, bringing more challenge for DeePSD to converge. The collision loss for DeePSD further generates noises and forces DeePSD to push clothes away from human model to achieve lower collision rates. In contrast, our LayersNet is able to faithfully rollout the garments even on this challenging case.

Table 8: We train LayersNet on our LAYERS and test on Cloth3D. We also compare DeePSD which is trained on Cloth3D. We re-sample the test samples from Cloth3D and include continuous sequences for better comparisons. The test set and training set on Cloth3D have no overlaps. Our LayersNet is able to achieve superior results on all scenarios.

| Method | T-shirt | Top | Trousers |
|---|---|---|---|
| DeePSD (trained on Cloth3D) | 36.69±15.34 | 26.52±9.15 | 29.86±13.47 |
| LayersNet (trained on LAYERS) | **23.84±15.81** | **14.17±5.92** | **23.16±11.38** |

| Method | Skirt | Jumpsuit | Dress |
|---|---|---|---|
| DeePSD (trained on Cloth3D) | 55.66±22.11 | 26.86±6.65 | 54.71±49.69 |
| LayersNet (trained on LAYERS) | **41.48±17.73** | **23.01±9.93** | **35.51±37.73** |

able to predict faithful rollouts. MGNet (Zhang et al., 2022) is able to generate 3D garments on training samples while struggles to generalize to unseen garments from test examples due to the garment-specific design. In original paper of MGNet, they train MGNet with only on 300 frames of data with the same garment topology, while in our dataset each garment is unique with different topology. In contrast, our LayersNet achieves faithful predictions in this simplified case.

On the other hand, as shown in Figure 5, the garments in our dataset, especially the outer clothes, are more flexible and are able to respond to various garment attributes, such as falling off the shoulder due to small frictions in this case. The high flexibility brings more challenges to DeePSD, leading to difficulties in convergence. The collision loss for DeePSD further introduces noises due to inaccurate garment meshes and forces DeePSD to push the clothes away from human to achieve lower collision rates. In contrast, our simulation-based LayersNet animates garments in topology-independent and unified manners and still achieves faithful rollouts even on such challenging case.

## A.2 MODEL GENERALIZATION

Since our LayersNet learns a topology-independent simulation model and is highly generalizable to unseen scenarios, we test LayersNet , which is trained on our dataset LAYERS, on Cloth3D

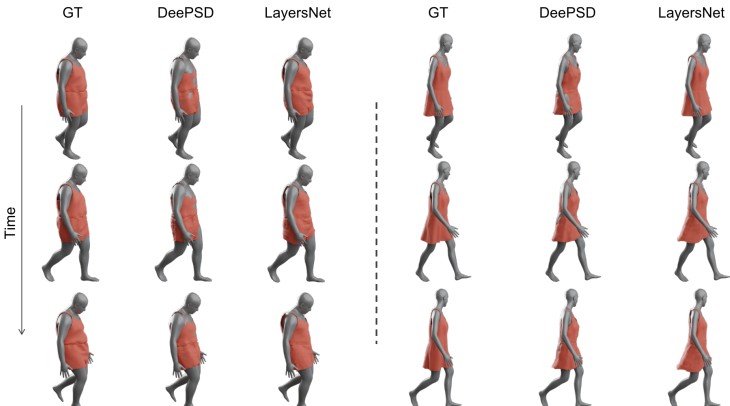

Figure 6: Samples are from Cloth3D Bertiche et al. (2020). The human model on the left wears a jumpsuit and is walking, while the human on the right wears a dress and is spinning. Our LayersNet achieves faithful rollouts. DeePSD is able to animate the garments with more stiff dynamics, while LayersNet can predict garments following physics laws, such as the inertia shown in the dress.

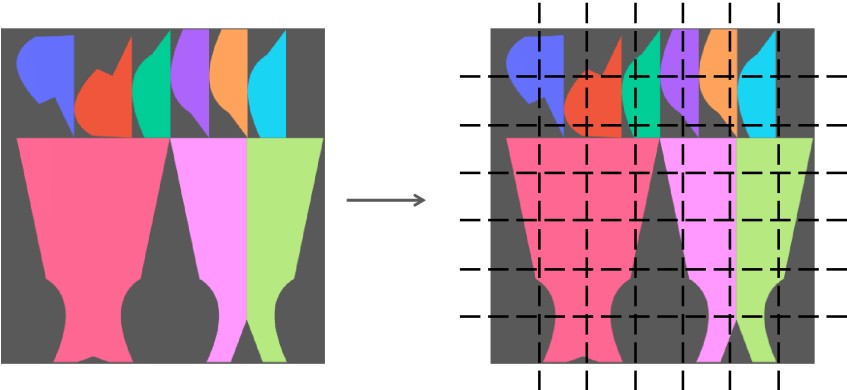

Figure 7: We grouped garment's vertices into patches given the corresponding UV mapping. Based on the coordinates in UV, we group the vertices according to their positions. Subsequently, we divide the 3D garment mesh into patches based on the groups.

(Bertiche et al., 2020) for generalization and compare the original DeePSD trained on Cloth3D. We re-sample the test set to include continuous sequence of samples. The test set and training set for training DeePSD on Cloth3D have no overlaps. As shown in Table 8, our LayersNet achieves superior performance on all types of garments. As shown in Figure 6, when test on unseen samples, predictions by DeePSD tend to be stiff while LayersNet still achieves faithful and vivid rollouts, suggesting the effectiveness of our simulation-based method which animates garments in a unified manner.

### A.3 IMPLEMENTATION DETAILS

**Patched Garment Model.** We group the particles in garment's mesh given corresponding UV mapping. Specifically, we divide the UV mapping into square patches according to the UV coordinates, as shown in Figure 7. Then, we group the garment's vertices in 3D space given the grouped UV mapping. When building the connections $\boldsymbol{E}^M$ of the patched garments, we connect patch $i$ and patch $j$ iff there is at least one pair of vertices within the patches are connected in 3D space.

**LayersNet details.** To obtain the world space edges $\boldsymbol{E}^W$, we adopt $R = 0.4$ to calculate the neighbors from the human mesh and $R = 0.6$ for different layers of garments. We adopt $h = 1$ for

Table 9: Euclidean error (mm) on sampled LAYERS with maximum sequence length of 35 frames. The collision rates between different layers of garments are shown under **L-Collision**, while the collision rates between garments and human bodies are shown under **H-Collision**. Models trained with collision loss are marked by +. When training DeePSD with collision loss, we extend Equation 14 and include the collisions between different layers of garments as extra loss term. In constrast, our model only applies the basic collision loss between garments and human bodies following Equation 14. Our LayersNet achieves superior results on all types of garments. Even without explicitly punishing collisions between layers of garments, LayersNet still achieves low collision rates among garments.

| Methods | Jacket | Jacket + Hood | Dress | Jumpsuit | Skirt |
|---|---|---|---|---|---|
| LayersNet | 706.3±558.5 | 588.2±460.5 | 436.3±343.7 | 289.3±194.4 | 322.9±58.9 |
| LayersNet+$\mathcal{L}_n$ | 717.6±609.8 | 577.5±458.1 | 448.2±452.2 | 277.3±293.1 | 274.6±94.4 |
| LayersNet+$\mathcal{L}_n, \mathcal{L}_c$(full) | 684.9±554.9 | 566.2±425.4 | 501.2±466.9 | 321.1±274.7 | 378.6±143.0 |

| Methods | Pants | T-shirt | Overall | L-Collision | H-Collision |
|---|---|---|---|---|---|
| LayersNet | 307.2±164.0 | 282.9±170.8 | 558.9±409.0 | 5.31%±3.88% | 16.41%±7.18% |
| LayersNet+$\mathcal{L}_n$ | 291.2±301.7 | 272.4±198.2 | 560.6±452.2 | 4.51%±2.98% | 10.01%±5.62% |
| LayersNet+$\mathcal{L}_n, \mathcal{L}_c$(full) | 349.3±238.4 | 331.7±226.9 | 567.2±432.8 | 4.94%±2.67% | 3.58%±2.83% |

the inputs of all objects' states. The hyperparameters $\lambda_m, \lambda_n, \lambda_c$ in our loss term are set to 1. We adopt Adam optimizer with an initial learning rate of 0.001 and a decreasing factor of 0.5 every two epochs. The batch size is set to 4.

We adopt three different encoders for meshes, garment attributes, and wind attributes. This is because the three components belong to different domain space and have different dimensions. Since the states of garments mesh and human bodies mesh are from the same domain, we share the encoder for them. All the encoders are two-layer MLPs with dimensions 128. The only difference is the input dimensions. We adopt 4 blocks of modified Transformer block in LayersNet, with hidden dimensions 128 for each block. The number of head in multi-head attention is set to 8, while 4 heads apply the attention mask generated by $\boldsymbol{E}^M$, and 4 heads adopt the attention mask generated by $\boldsymbol{E}^W$. For the decoder, we adopt a three-layer MLPs with a forward dimension of 128 and an output dimension of 3. When concatenating the nearest point on human mesh in Equation 10, we mask the point $\boldsymbol{v}_{h,i}^{t+1}$ to zeros if the there is no edge $e_{h,i} \in \boldsymbol{E}^W$ connecting the point $\boldsymbol{v}_{h,i}^{t+1}$ and garment point $\boldsymbol{v}_i^{t+1}$. For the inputs of garment mesh and human body mesh, we adopt relative positions to the root of human body mesh. Since the wind's attributes are still measured in global coordinates, we also convert them to the relative coordinates in implicit manner, i.e. convert the value of strength in global coordinates to local coordinates defined by the root of human body. Specifically, we concatenate the position, velocity, and acceleration of the human body's root point $\boldsymbol{v}_r^t$ as extra features $\boldsymbol{w}^t = \{\boldsymbol{q}^t, s^t, \boldsymbol{v}_r^t\}$ and let LayersNet learns the wind's features in relative coordinates. We apply $\Delta t = 1$ in our experiments, which is independent from the real time interval between each frames, which is 0.33s. When training LayersNet, we normalize the meshes' states across the whole training set before feeding into the model, which is a commonly adopted processing in literature.

## A.4 ABLATION STUDY ON LAYERSNET

In this section, we analyze our model in the following aspects: (a). vanilla LayersNet; (b). LayersNet with normal loss; (c). LayersNet with both normal loss and collision loss.

As shown in Table 9, the vanilla LayersNet achieves low Euclidean errors on all types of garments, suggesting the effectiveness of our simulation-based method. With the normal loss term $\mathcal{L}_n$, which aims to smooth the surface of garments, achieves lower collision rates in terms of both body-to-cloth penetrations and collisions between different layers of garments, leading to more faithful rollouts comparing with vanilla LayersNet. When training with both normal loss and collision loss, LayersNet further reduce the body-to-cloth penetrations. Though our complete version of LayersNet has slightly higher Euclidean errors, the lower collision rates lead to more convincing and faithful results. A good example is shown in Figure 3 of the main text.

## A.5 FUTURE WORK

In this work, we propose a novel multi-layered 3D dataset and make the first attempt to animate garments though simulation pipeline which achieve superior performance in both LAYERS and other generalization scenarios. Since we regard all objects as particles, our model is able to animate various garments with different typologies driven by different types of outer forces, such as human

bodies and wind. However, since the predictions of future frames are based on previous rollouts by the model, the errors accumulate as the length of predictions increases, which is a common problem in simulation. In this work, we train the model and alleviate the problem by predicting two continuous frames as shown in Equation 16. We will explore more strategies, such as adding noise to input data and forcing the model to learn to correct errors, to enable the model to rollout long sequences while keeping the accumulated errors relatively small at the same time.

