# OpenReview forum: "Multi-Layered 3D Garments Animation"
_ICLR.cc/2023/Conference — Submitted to ICLR 2023_

### Official Review · Reviewer_X85t · 2022-10-23

**Confidence:** 4
**Clarity, Quality, Novelty And Reproducibility:** See above.
**Correctness:** 3
**Technical Novelty And Significance:** 3
**Empirical Novelty And Significance:** 3
**Recommendation:** 6

**Strength And Weaknesses:**

Strengths:
1. Multi-layered cloth motion dataset is indeed in great need. This dataset can boost the advances of learning-based garment estimation and animation research.
2. Results show that LayersNet has a better performance than previous work in terms of predicting multi-layer garment motion.

Weaknesses:
1. Missing references:
 - Physics Simulation by Neural Network: Liang, Junbang, Ming Lin, and Vladlen Koltun. "Differentiable cloth simulation for inverse problems." Advances in Neural Information Processing Systems 32 (2019).
 - Data-driven Cloth Model: Shen, Yu, Junbang Liang, and Ming C. Lin. "Gan-based garment generation using sewing pattern images." European Conference on Computer Vision. Springer, Cham, 2020.
2. In Table 3, is there any reason why the LayersNet is not trained and tested with inner garments only as well? Otherwise, I think DeePSD for inner garments only is not needed either.
3. Does the wind velocity affect the accuracy of the model? How well does the model perform with different wind speeds? In the simulation, very minor differences in the initial setting can cause significantly different end results (i.e., known as instability of the simulation). Does that happen as well for the model? If so, is there any particular way to overcome this so that the prediction errors can make more sense? Please add some discussions regarding this topic.
4. In Table 4, the proposed method does not perform as well as previous works in regards to L-Collision and H-Collision. The authors should analyze and explain the phenomena but I didn't see any in the paper.

**Summary Of The Paper:**

The paper proposes a new dataset of garment motions under different human movements as well as wind forces. It also introduces a new network that can predict garment motions of different layers at the same time.

**Summary Of The Review:**

Although the dataset is a good contribution, the proposed method has some clarity issues that prevent this paper from meeting the bar. I would be happy to change my score as long as the issues above are addressed.

---

> ### Author Response · Authors · 2022-11-17
> **Author Response to Reviewer X85t [1/1]**
>
> We thank reviewer for appreciating the multi-layered garment dataset and the simulation-based method. We have revised the paper to reflect the feedback, where revision is marked with magenta color. Below we address each concern in detail.
>
> ## Q1: Missing reference
> Since our main contributions are the synthetic 3D dynamic multi-layered dataset and data-driven method to animate 3D garments, we mainly compare existing synthetic datasets and related methods.
> We also add them to our related work.
>
> ## Q2: Why only DeePSD is trained and tested within inner garments?
> As shown in Table 3, we compare DeePSD trained with inner garments and multi-layered garments to show the challenges brought by multi-layered garments. DeePSD with only inner garments also indicates that our implementation of DeePSD is valid. We explain the reason for conducting experiments with inner garments in paragraph 2 Section 5.2.
> We also train LayersNet with only inner garments as shown below. Another baseline, MGNet, from Motion Guided Deep Dynamic 3D Garments 2022, is also included. Since MGNet is specially designed for single-layered garments, we test it only with inner garments.
>
> | Methods              | Tight (T) | Tight+Wind (T+W)  | Loose (L)  |  Loose+Wind (L+W)  |
> | --------------------- | ------------------ | ------------------ | ------------------ | ------------------ |
> | DeePSD*(inner garments only)    | $225.3\pm106.4$ | $239.5\pm103.9$ | $501.3\pm300.1$ | $577.5\pm373.9$ |
> | MGNet*(inner garments only)     | $5219.2\pm1565.8$ | $5186.8\pm1754.8$ | $4432.7\pm1438.0$ | $4595.0\pm1215.2$ |
> | LayersNet*(inner garments only) | $260.3\pm254.4$ | $278.6\pm328.6$ | $378.0\pm293.0$ | $363.6\pm311.4$ |
>
> To generate the scenarios with only inner garments,
> we simply remove the outer garments in corresponding sequences.
> However,
> since the inner garments interact with outer garments during simulation,
> the dynamics of inner garments are influenced by outer garments.
> Consequently,
> the inner garments themselves are slightly inconsistent with physics laws.
> Since DeePSD does not consider the physics consistency,
> it is less affected and achieves slightly better results on split T and T+W.
> MGNet has difficulties in generalizing to garments with different topologies. Please refer to Section A.1 in Appendix for more details and qualitative comparisons.
> Our LayersNet achieves lower errors especially on loose garments,
> suggesting the effectiveness and higher generalization abilities of animating loose garments. To further demonstrate the generalization abilities of LayersNet, we train our model on LAYERS and test it on Cloth3D. More qualitative and quantitative results can be found in Section A.2 in Appendix.
>
> ## Q3: Does the wind velocity affect the accuracy of the model? What about the problem of instability of simulation?
> As shown in Table 3, existing methods are affected by wind more obviously and have worse performance. In contrast, our method is more stable and gains improvements through the variations introduced by the wind.
>
> The LayersNet is also slightly affected by accumulated errors.
> However,
> LayersNet still achieves faithful rollout with a maximum sequence length of 35 frames on unseen scenarios, including different human bodies, wind, garment topologies and attributes.
> In this work,
> we also alleviate the exposure bias by predicting two continuous frames during training.
> We will explore more strategies in the future,
> such as adding noise to input data and forcing the model to learn to correct errors,
> to enable the model to rollout longer sequences
> while keeping the accumulated errors relatively small at the same time.
>
> We discuss and analyze the limitations in Section A.5 in Appendix.
>
> ## Q4: Proposed method has higher collision rates
> We briefly explain it at the former part of paragraph 2 in Section 5.3.
> Without explicit collision handling between garments, our model is able to make a balanced trade-off between the faithfulness of predictions and collision rates. DeePSD with specifically designed collision loss achieves lower collision rates but much higher errors. The main reason that DeePSD has lower collision rates is that the model has difficulties in convergence and pushes garment meshes away from the human body. We add some visualization results for DeePSD in Section A.1 Appendix.

---

> > ### Comment · Reviewer_X85t · 2022-11-22
> > **Thanks**
> >
> > Thank you for the responses. My concerns are all well addressed. I will change my score to weak accept.

---

> > > ### Author Response · Authors · 2022-11-25
> > > **Author Response to Reviewer X85t**
> > >
> > > We thank the reviewer for your valuable time and positive comments on our work.

---

### Official Review · Reviewer_g4UX · 2022-10-24

**Confidence:** 4
**Correctness:** 3
**Technical Novelty And Significance:** 2
**Empirical Novelty And Significance:** 2
**Recommendation:** 5

**Clarity, Quality, Novelty And Reproducibility:**

The paper is well written and I could comfortably follow it. The work should also be reproducible since authors promise to release code. Will the authors also release the simulation code? This could be very useful for the community.The originality seems a bit limited to me as authors primarily rely on existing work TIE, Shao 2020 to handle garment  modelling (see weakness section). It will help if authors address this point as it is one of the main contributions of the work.

**Details Of Ethics Concerns:**

I do not see any immediate ethical concerns arising out of the work.

**Strength And Weaknesses:**

Strengths:
+ The task of modelling multi-layered garments is quite challenging and not a lot of works address this.
+ Typical works on garment modelling on human body do not consider external factors such as wind, which is interesting in this work.
+ Authors chose particle based physics modelling for the garments which is topology independent and allows modelling external factors like wind as wind can be treated as a set of particles.
+ The authors will release the dataset and code.

Weaknesses:
- Technical novelty seems limited, as the garment modelling is handled by TIE, Shao et al. 2020. Can the author clarify the differences between the approaches.

- In eq. 3 the garment is modelled as a function of body and previous state of the garment. Since the paper primarily deals with multi-layered garments, should't the garment also be a function of other garments on the body?
eg: let's say a person is wearing a loose skirt and a long jacket on top, such that the bottom of the jacket draper around the top of the skirt. Isn't the jacket deformation more dependent on the skirt than the body?

- Eq. 14 only handles collisions between the body and the garments, but since authors deal with multi-layered garments, what about the collisions between different layers of garments?

- Related work on multi-layered garment is completely missing. This important as multi-layered garments are the main contribution of the work. Authors should add a paragraph about it as currently it appears that the authors are the first one to address multi-layered garments which is not true.
--> SMPLicit, CVPR'21 (not cited) has shown generation of multi-layered garments.
--> SimulCap, CVPR'19 (not cited) Also use multi-layered garments + physics based simulation for motion capture.
--> Another concurrent work: Layered-Garment Net, ACCV'22 (not cited) also generates multi-layered clothing using simulated data.

Clarifications:
- Eq. 12: Shouldn't this be a double summation. One over multi-layered garments and other over vertices of the garment.
- What is f(.) in the line just above eq. 3?
- There are existing works like TailorNet, Patel et al. 2021 that also use PBS to drape garments on human body. Can the authors also provide details on how simulating multiple layers is more challenging than simulating single layered clothing?


**Summary Of The Paper:**

Authors address the task of simulating multi-layered 3D garments on a human body. Authors fist curate a dataset of multi-layered garments under various poses and wind effects and use this data to train LayersNet that predicts garment deformations as a function of body shape and previous garment state.

**Summary Of The Review:**

I like the task and formulation that authors propose in this work. It is not clear to me how is the existing garment modelling different than the particle based modelling in TIE, Shao 2020. Moreover, parts of the formulation seem to be tailored for modelling single layered garments and not multiple layers. eg: intersection is handled only with the body and garment and not garment-garment.

Can the authors also provide video results from their method. This is important as the authors deal with garment simulation over time and we should see the quality of the simulated data (the proposed dataset) as well as the results from LayersNet. With provided results it is difficult to access whether the clothing deforms smoothly or not.

---

> ### Author Response · Authors · 2022-11-17
> **Author Response to Reviewer g4UX [2/2]**
>
> ## Q4: f(.)
> Thanks for the clarification. This should be a function to split garments into patches. We correct this symbol.
>
> ## Q5: Why are multi-layered garments more challenging than single-layered
> As shown in Table 3, DeePSD performs much worse on multi-layered settings than single-layered garments, suggesting the difficulties in converging on multi-layered garments settings. First, the interactions between garments are more complex than single-layered garments. Second, the outer garments have more freedom to deform, such as slipping off the shoulders due to small frictions, leading to more interactions with inner garments and more challenges to learn.
>
> Since most existing datasets for garment animations are limited to single-layered garments driven only by human bodies,
> previous methods tend to model garments as functions of human body parameters
> and are not feasible in real scenarios,
> leaving a gap between experimental environments and real scenarios.
> We make an attempt to close the gap by the proposed dataset which includes more general settings close to realistic scenarios, such as multi-layered garments driven by the human body and wind.
> LAYERS also aims to encourage researchers to explore more general methods to animate garments.
> In addition, we propose LayersNet to animate garments in a unified manner by simulation-based pipeline.
>
> ## Q6: Releasing the simulation code
> Thanks for the suggestions, we will consider releasing the code for data generation.
>
> ## Q7: Video
> Thanks for the suggestions. As suggested, some videos are provided as  supplementary material.

---

> > ### Comment · Reviewer_g4UX · 2022-11-28
> > **Video results**
> >
> > Thanks for the video. The dataset looks high quality but I see a lot of artefacts in the predicted results? Some baseline needs to be provided as a reference to access the quality of the proposed approach (even if it is not a fully fair comparison).

---

> > > ### Author Response · Authors · 2022-12-01
> > > **Author Response to g4UX**
> > >
> > > We discuss the limitations of our LayersNet in A.5 in Appendix. Specifically, the LayersNet is slightly affected by accumulated errors. However, LayersNet still achieves faithful rollout with a maximum sequence length of 35 frames on unseen scenarios, including different human bodies, wind, garment topologies and attributes. In this work, we also alleviate the exposure bias by predicting two continuous frames during training. We will explore more strategies in the future, such as adding noise to input data and forcing the model to learn to correct errors, to enable the model to rollout longer sequences while keeping the accumulated errors relatively small at the same time.
> > >
> > > We include the rendered results for baselines in Appendix,
> > > such as the inner garments results in Figure 4, the complex multi-layered garments results in Figure 5, and the generalization results on Cloth3D in Figure 6.

---

> ### Author Response · Authors · 2022-11-17
> **Author Response to Reviewer g4UX [1/2]**
>
> We thank reviewer for appreciating the challenging dataset and the proposed highly generalizable method. We have revised the paper to reflect the feedback, where revision is marked with magenta color. Below we address each concern in detail.
>
> ## Q1: Technical novelty
> Our main contributions are two folded:
> 1. We propose a DYNAMIC MULTI-LAYERED 3D garment dataset, where different layers of garments are able to interact with each other constrained by physics laws and we introduce WIND as extra outer forces besides human body. Besides, each garment has not only unique topology, but also unique attributes, such as stiffness and friction
> 2. We make the first attempt to animate garments by any kind of driving factors, which are human bodies and wind in this paper, from the view of simulation by data-driven method.
>
> Simply applying TIE is insufficient to animate garments, since each scenario contains meshes with 10k to 30k vertices. To efficiently and faithfully animate 3D garments, we adopt the following adaptations:
> 1. We first split garments into patches as basic simulation units. In this way, our model only need to simulate less than 1000 patch-level units instead of 30k vertices as shown in equation 3.
> 2. To reconstruct the dynamics of vertices in each patch, we design a decoder to interpolate each vertex given the patch and nearest human body vertex. The first two adaptations mentioned above make our model learn topology-independent garment dynamics, leading to high generalization abilities as shown in Table 4.
> 3. We apply two attention masks to accelerate the convergence of LayersNet: one mask indicates the neighbors in global coordinates by $E^W$, the other suggests the connectiveness within the meshes by $E^M$. While $E^W$ enables model to be aware of the neighbors for collisions, $E^M$ indicates how the energy transits within the mesh manifold.
>
> ## Q2: Garments should be functions of other garments due to multi-layer; Explicit collision loss between layers of garments; Garment-wise MSE loss function;  Intersection is handled only with the body and garment and not garment-garment.
> It is worth noticing that we do not have the concept of "garment".
> Instead,
> our method aims to predict the dynamics of patches,
> which are the basic components of garments and the units for simulation.
> All interactions between patches, such as collisions, are considered and modeled in a unified manner by our method.
> Specifically,
> the attention masks indicated by $E^W$ deal with the neighbor patches including patches from different layers,
> while masks indicated by $E^M$ focus on the interactions within the mesh manifold.
>
> Thus,
> the patches are the main objects we deal with.
> Multi-layer-specific designs and explicit collision handling between garments are not necessary, which is one of the advantages of our method.
> Explicit collision loss between human and garments is adopted since human body is a driving factor and is not part of the predictions.
> The MSE loss is applied to all particles, where patches are also regarded as particles, in a unified manner.
>
> As mentioned in Q1, one of our contributions is to adopt simulation to animate garments, which provides a unified way to animate different layers of garments driven by any kind of outer factors and is easy to generalize.
>
>
> ## Q3: Related work about multi-layer garments
> Thanks for reminding of the missing citations. We add the potential papers in our related work.
>
> Since our main contributions are the synthetic 3D dynamic multi-layered dataset and data-driven method to animate 3D garments, we mainly compare existing synthetic datasets and related methods.
>
> Some existing datasets or work include the concept of multi-layered garments. However, they either refer to multiple separate garments mesh, such as upper shirt and lower pants with few overlappings, as multi-layered garments, or generate multi-layered garments that do not follow physics laws by forcing penetrated vertices out of inner garments.
> Instead, in our dataset, different layers of garments highly overlap and interact with each other constrained by physics laws.
>
> Since our dataset includes the 3D information, such as coordinates, we can easily generalize our dataset to other multi-layered garments settings. In contrast, no existing dataset can convert to our settings, such as 3D separate garment meshes with dynamic deformations and interactions between different layers of garments.

---

> > ### Comment · Reviewer_g4UX · 2022-11-28
> > **Technical novelty still not clear**
> >
> > Thanks authors for the detailed response. Q2 and 5 were very useful to me.
> > I agree the paper addresses a new problem setting ( Our main... by data-driven method)
> > I'm still not convinced by points 1,2,3 (We first split garments .... the mesh manifold.)
> >
> > Regarding 1, 2: The practicing of dividing high resolution meshes into patches/ low res. meshes, for simulation, prediction etc. is quite well known in computer graphics. Do the authors mean that they use a neural network for this encoding/ decoding and this is the novel part?
> >
> > Regarding 3: My understanding is that Pfaff et al., 2021, proposed the usage of the world and mesh connectivity for predicting future states in of a simulation. How is the proposed formulation different here (Sec 4.1)?
> >
> > The proposed method can be seen as: 1) Map mesh to patches (Ma et. al, 2021) 2) Use TIE, Shao et al. 2020 to simulate patches 3) Train a NN to decode patches back to mesh. Part 3) seems most novel here and this seems like a limited contribution.

---

> > > ### Author Response · Authors · 2022-12-01
> > > **Author Response to g4UX**
> > >
> > > Thanks for your reply. It's worth noting that our contribution consists of both the new challenging dataset (beyond existing commonly used ones such as Cloth3D) and the proposed method.
> > >
> > > To be specific, our technical novelty lies in **the first attempt**
> > > to propose a simulation-inspired method,
> > > which rollouts the future dynamics based on previous predictions,
> > > for **data-driven garment animation**.
> > > As pointed out in the paper, existing methods for data-driven garment animation often regard garmants as functions of human.
> > > Through the carefully established dataset and the proposed method,
> > > we reveal the limitation of existing methods and their design principle.
> > > Moreover,
> > > we demonstrate the great potential of simulation-inspired design principles in this task,
> > > which are of high quality, general, efficient and flexible.
> > > While we hope our dataset and method can attract more research attention towards this direction,
> > > there are certainly plenty of rooms for improvements in the method modules,
> > > which, however, is not the main focus of this paper.
> > >
> > > We will revise our paper to emphasize our main points.

---

> ### Author Response · Authors · 2022-11-26
> **Looking Forward to Your Reply**
>
> We thank the reviewer again for your valuable time and insightful comments. We hope the updated manuscript and our rebuttal text can address all your concerns and look forward to your reply. We are happy to answer any follow-up questions that are currently affecting your review. Thank you for your time!

---

### Official Review · Reviewer_9oGC · 2022-10-24

**Confidence:** 4
**Correctness:** 4
**Technical Novelty And Significance:** 3
**Empirical Novelty And Significance:** 3
**Recommendation:** 6

**Clarity, Quality, Novelty And Reproducibility:**

The data-driven motivation is not particularly clear to me.
- In this work, the proposed method learns to mimic a physical simulator’s behavior. The author needs to justify its use cases, i.e., “why data-driven method?” Given that we have options for physical-based clothes animation solutions, I expect the author to discuss when and why data-driven is beneficial.
- I think the author can answer the question at least from one of the three aspects: 1) the data-driven method is faster than the Blender’s simulator; 2) the data-driven simulator is better than a real-time physical-based simulator (e.g., 3) data-driven simulator can offer other capacities (e.g., system identification, where end-to-end differentiability allows inferring physical params through inverse problem solver; or sim2real, where the learned particle-dynamic model can be applied for real-world data).
- However, I cannot find a runtime report or comparison against a real-time garment simulator or a demonstration of downstream demonstrations.

Dataset details:
- It’s also unclear how fast Blender’s clothes simulation is and what method they use (it is also particle based?).
- Training/testing splits -- during testing how many samples are from unseen poses / unseen human shapes/ unseen clothes materials?

Overall the comparison experiments are not entirely thorough.
- First, it should be compared against a differentiable particle-based physical solver, if possible, such as DiffCloth, or a real-time physical solver for a speed-performance trade-off.
- Another data-driven method, SCALE (Ma et al.), should also be compared. SCALE might not be directly modeling wind. But the author can also add wind as an additional attribute, similar to what the author did for DeePSD.
- Besides, the paper also lacks qualitative comparison against DeePSD. It seems the performance gap is quite large quantitatively, but I would also like to know how it works visually compared to DeePSD.


**Strength And Weaknesses:**

Pros:
- The new capacity for learning-based clothes simulation to handle both wind-cloth, body-cloth, and inter-cloth interactions.
- Superior performance compared against DeePSD through a thoughtful exp design.
- The paper is clearly presented. Equations/tables/figures are all easy to parse. The intro is clear. Related covers most of the relevant works.
- New large-scale dataset with realistic simulation and unique designs (e.g., wind).

Cons:
- The problem setting of data-driven animation here is not well-motivated.
- The comparison study is somewhat limited.
- Lack of qualitative demonstrations and comparisons.

Mixed:
- Technical contribution. The paper belongs to a novel application / resemble of existing techniques (1. data-driven particle-based simulation from Shao et al. and 2. patch-based garments representation used in Ma et al. and earlier works. e.g., Kavan et al. Siggraph 11, Feng et al. 2010)
- The above two core contributions the author claimed to differentiate the proposed work from prior works are not entirely original. That said, I still appreciate the novel ensemble and applications to data-driven simulation under the wind.



**Summary Of The Paper:**

MULTI-LAYERED 3D GARMENTS ANIMATION

This paper presents a novel deep learning-based 3D garments animation method. The method learns to model interactions between 1) cloth-body, 2) multiple cloth layers 3) wind and clothes. Specifically, the algorithm decomposes the clothes into connected patches and evaluates the deformation based on a learning-based particle physics simulator (Shao et al., 2022). A new simulation dataset has been proposed to study this new setting.



**Summary Of The Review:**

Overall I am on the fence. I would like to see how the author addresses my aforementioned concerns and questions.


----------------------------------------------
Post-rebuttal.

The authors addressed most concerns by providing additional experiments and revised the submission accordingly.

Overall, it is a weak-accept to borderline-ish submission.

The paper presentation is quite clear, and the technical approach is straightforward, as pointed out by both the authors and reviewers -- fusing particle simulation in TIE (Shao et al.) with patch-based representation in SCALE (Ma et al.) for efficiency. The method achieves the SOTA in runtime and speed and can handle external force and unseen clothes types.

It is a well-executed and well-presented paper after the revision. The proposed combination is somewhat new and, indeed, solves practical challenges of prior arts (efficiency, external force, and multi-layer interaction). It's not a ground-breaking or surprising idea, but it sets a new state for the field and brings new capacity. Hence it doesn't harm to get accepted for its practicality.

---

> ### Author Response · Authors · 2022-11-17
> **Author Response to Reviewer 9oGC [3/3]**
>
> ## Q4 (Technical) Contributions
> Our main contributions are two folded:
> 1. We propose a DYNAMIC MULTI-LAYERED 3D garment dataset, where different layers of garments are able to interact with each other by physics laws and we introduce WIND as extra driving factors besides the human body. Our dataset includes various physics parameters, such as friction and stiffness, which can be applied to different tasks, such as physics parameters estimation.
> Most existing datasets for garment animations are restricted to human bodies with single-layered garments.
> Though some existing work tries to extend their methods to animate multi-layered garments, the inner and outer garments do not follow physics laws,
> such as building outer garments by regarding the inner garments as fixed surfaces,
> leading to a gap between experimental environments and real scenarios.
> Methods developed on top of these datasets tend to model garments as functions of human body parameters.
> We thus propose our dataset to close the gap between experimental and realistic cases by providing multi-layered garments, which follow physics laws, and wind as extra driving factors.
> Our dataset also encourages the community to develop more general approaches to animate garments besides the human body models.
> 1. We make the first attempt to animate garments by any kind of driving factors, which are human bodies and wind in this paper, from the view of simulation by data-driven method. We only adopt the similar notion of "patch" in SCALE, which is regarded as a two-level hierarchy particle in our method and shares nothing more similar. Moreover, simply applying TIE is insufficient to animate garments, since each scenario contains meshes with 10k to 30k vertices.
> To efficiently and faithfully animate 3D garments, we adopt the following task specific adaptations: 1. We first split garments into patches as basic simulation units. In this way, our model only needs to simulate less than 1000 patch-level units instead of 30k vertices as shown in equation 3. 2. To reconstruct the dynamics of vertices in each patch, we design a decoder to interpolate each vertex given the patch and nearest human body vertex. The first two adaptations mentioned above make our model learn topology-independent garment dynamics, leading to high generalization abilities as shown in Table 4. 3. We apply two attention masks to accelerate the convergence of LayersNet: one mask indicates the neighbors in global coordinates by $E^W$, the other indicates the connectiveness within the meshes by $E^M$. While $E^W$ enables the model to be aware of the neighbors for collisions, $E^M$ indicates how the energy transits within the mesh manifold.

---

> ### Author Response · Authors · 2022-11-17
> **Author Response to Reviewer 9oGC [2/3]**
>
> ## Q3 Comparison study is limited
> ### Physics-based solver comparison
> Since we focus on data-driven methods, comparison experiments are conducted among data-driven methods only. While Blender adopts physics-based methods to simulate garments, we compare the inference time per frame mentioned in Q2, where our model is 40 times faster than Blender.
> Moveover, physical solver, such as DiffCloth, usually solves the simulation as an optimization problem. Since it adopts implicit metrics to measure the performance, it cannot compare with data-driven methods directly in terms of the L2 distance per vertex.
>
> ### SCALE
> SCALE aims to reconstruct the mesh of garments in specific resolutions given single-layered scans of clothed body and minimum clothed body, which has different settings from our garment animation. Moreover, SCALE is a garment-specific model which is hard to generalize to different topologies even if the garments belong to the same categories. Thus, it is not suitable to be applied on LAYERS where different sequences have different garments with completely different topologies.
>
> Instead,
> we compare another dynamic garment animation model in Motion Guided Deep Dynamic 3D Garments 2022, which we refer it as MGNet. MGNet is a dynamic model which considers continuous deformations brought by the human body along the temporal axis. It shares some similarities to SCALE. For example, MGNet adopts the local coordinates for each vertex, which has a similar design to the local elements in SCALE. Since it is designed specifically for single-layered garments, we compare MGNet with only inner garments on the four different splits.
>
> | Methods              | Tight (T) | Tight+Wind (T+W)  | Loose (L)  |  Loose+Wind (L+W)  |
> | --------------------- | ------------------ | ------------------ | ------------------ | ------------------ |
> | DeePSD*(inner garments only)    | $225.3\pm106.4$ | $239.5\pm103.9$ | $501.3\pm300.1$ | $577.5\pm373.9$ |
> | MGNet*(inner garments only)     | $5219.2\pm1565.8$ | $5186.8\pm1754.8$ | $4432.7\pm1438.0$ | $4595.0\pm1215.2$ |
> | LayersNet*(inner garments only) | $260.3\pm254.4$ | $278.6\pm328.6$ | $378.0\pm293.0$ | $363.6\pm311.4$ |
>
> To generate the scenarios with only inner garments,
> we simply remove the outer garments in corresponding sequences.
> However,
> since the inner garments interact with outer garments during simulation,
> the dynamics of inner garments are influenced by outer garments.
> Consequently,
> the inner garments themselves are slightly inconsistent with physics laws.
> Since DeePSD does not consider the physics consistency,
> it is less affected and achieves slightly better results on split T and T+W.
> Our LayersNet achieves lower errors especially on loose garments,
> suggesting the effectiveness and higher generalization abilities of animating loose garments.
> MGNet has difficulties in generalizing to garments with different topologies. Please refer to Section A.1 in Appendix for more details and qualitative comparisons.
>
> ### Qualitative results from DeePSD
> We add some visual results in Section A.1 in Appendix.
> The main reason that DeePSD has difficulties in convergence is that
> garments in our dataset are more flexible.
> As shown in the qualitative comparisons in Section A.1 in Appendix,
> some of the outer garments in our dataset will fall off the shoulder
> and deform a lot due to the motion of the human and wind,
> which is also an advantage of our dataset that our garments are not stiff
> and are controlled by various attributes such as friction and stiffness.
> Though the human poses may be similar among different frames,
> the dynamics of the garments are different,
> bringing difficulties to DeePSD which models garments as a function of human poses and shapes.
> Since the basic DeePSD struggles to predict reasonable outfits,
> the collision loss introduces noises
> that makes the model push garments away from the body to achieve lower collision rates.

---

> ### Author Response · Authors · 2022-11-17
> **Author Response to Reviewer 9oGC [1/3]**
>
> We thank reviewer for appreciating the multi-layered garment dataset and the simulation-based method. We have revised the paper to reflect the feedback, where revision is marked with magenta color. Below we address each concern in detail.
>
> ## Q1: Data-driven motivation is not clear
> Our data-driven method has advantages over physics-based simulator (Blender) in the following aspects:
>
> | Methods               | Time Per Frame(s)  |
> | --------------------- | ------------------ |
> | Blender               | $10.31$            |
> | LayersNet             | $0.24$             |
>
> 1. The inference speed per frame of our data-driven method is around 40 times faster than Blender.
> 2. To get the observation data, we choose the physics-based simulator as a tool to generate multi-layered garments, which are hard to capture in realistic data. As long as we can observe, we are able to learn the dynamics regardless of the potential physics properties. If the physics rules and formula for clothes were unknown, it's hard for physics simulator to work. On the contrary, if we can observe how the garments deform, our data-driven method can still learn the objects' dynamics. Thus, the effectiveness of our data-driven methods is not influenced by the changes of real physics properties.
> 3. Data-driven methods are able to generalize to different materials of objects in a unified manner. Physics-based methods have to adopt specific physics formulas to different materials, such as fluid and rigid objects. Data-driven method is able to control the physics properties by feeding an attribute vector into the neural network. In our case, as shown in the following table, the LayersNet trained on LAYERS also achieves faithful performance when testing on Cloth3D, which contains different garments with different attributes. Notice that the test set on Cloth3D is resampled to get sequences of data with a maximum length of 40 frames.
>
> | Methods               | T-Shirt | Top | Trousers | Skirt | Jumpsuit | Dress | Total |
> | --------------------- | ------------------ | ------------------ | ------------------ | ------------------ | ------------------ | ------------------ | ------------------ |
> | DeePSD (trained on Cloth3D)   | $36.7\pm15.3$ | $26.5\pm9.1$ | $29.9\pm13.5$ | $55.7\pm22.1$ | $26.9\pm6.6$ | $54.7\pm49.7$ | $37.7\pm31.2$ |
> | LayersNet (trained on LAYERS) | $23.8\pm15.8$ | $14.2\pm5.9$ | $23.2\pm11.4$ | $41.5\pm17.7$ | $23.0\pm9.9$ | $37.7\pm25.3$ | $28.0\pm17.9$ |
>
> 4. Data-driven methods, which are fully differentiable, such as our model, are easy to apply to inverse problem, such as physics parameters estimation and motion control.
>
> ## Q2: Dataset details
> 1. Cloth simulation in Blender is physics-based models such as Mezger's Cloth Collision Model (Improved Collision Detection and Response Techniques for Cloth Animation, [pdf](https://publikationen.uni-tuebingen.de/xmlui/handle/10900/48378)) . Internally, cloth physics is simulated with virtual springs that connect the vertices of a mesh. Our data-driven model is 40 times faster than Blender. The inference time for Blender is calculated during data generation.
>
> | Methods               | Average Time Per Frame(s)  |
> | --------------------- | ------------------ |
> | Blender               | $10.31$            |
> | LayersNet             | $0.24$             |
>
> 2. The samples from the training set, validation set, and test set are completely mutual exclusive, such as the human body shape, body motions, garment attributes, and garments topologies. Please refer to Section 3 for more details.

---

> ### Author Response · Authors · 2022-11-26
> **Looking Forward to Your Reply**
>
> We thank the reviewer again for your valuable time and insightful comments. We hope the updated manuscript and our rebuttal text can address all your concerns and look forward to your reply. We are happy to answer any follow-up questions that are currently affecting your review. Thank you for your time!

---

> > ### Comment · Reviewer_9oGC · 2022-11-27
> > **Thank you**
> >
> > Most questions are addressed. Thank you.
> >
> > Please revise the paper and motivate the data-driven simulation better. Runtime efficiency and ability to generalize can be a strong argument (I am not 100% convinced by your bullet points 2 and 3 BTW. As mentioned in your rebuttal, one can hardly capture realistic deformation data, then most likely observation data is still from physical simulation and we can have access to its true dynamics; hence it's more like an efficient data-driven solution mimicking time-confusing physical simulation unless you provided evidence that dynamics can be learned from real-world data. Regarding controllability, there's no evidence that the proposed model is controllable through the attribute vector).

---

> > > ### Author Response · Authors · 2022-12-06
> > > **Author Response to Reviewer 9oGC [2/2]**
> > >
> > > ## Q2: Controllablity
> > > Our model is able to control the attributes of garments,
> > > such as the stiffness,
> > > through the attribute vectors.
> > > We visualize some samples [**HERE**](https://youtu.be/fNDuJbaBF5Q).
> > >
> > > | L2 Errors (mm)  | Jacket      | Jacket + Hood        | Dress      | Jumpsuit       |
> > > | ----------------| ----------------| --------------- | ----------------| --------------- |
> > > | LayersNet (zero attr)  | 690.3 $\pm$ 547.3 | 577.2 $\pm$ 434.4 | 510.4 $\pm$ 462.9 | 327.2 $\pm$ 271.5 |
> > > | LayersNet       | **684.9 $\pm$ 554.9** | **566.2 $\pm$ 425.4** | **501.2 $\pm$ 466.9** | **321.1 $\pm$ 274.7** |
> > >
> > > | L2 Errors (mm)  | Skirt      | Pants        | T-shirt      | Overall       |
> > > | ----------------| ----------------| --------------- | ----------------| --------------- |
> > > | LayersNet (zero attr)  | 386.3 $\pm$ 129.1  | 382.1 $\pm$ 281.6 | 341.3 $\pm$ 255.7 | 575.6 $\pm$ 431.6 |
> > > | LayersNet       | **378.6 $\pm$ 143.0** | **349.3 $\pm$ 238.4** | **331.7 $\pm$ 226.9** | **567.2 $\pm$ 432.8** |
> > >
> > > We also report the quantitative results showing the controllablity of the garment attribute vectors in our LayersNet.
> > > Our model with constant zero vectors as attributes is indicated in the Table.
> > > As shown in the Table,
> > > our model with meaningful garment attribute vectors achieves lower errors,
> > > suggesting the effectiveness and controllablity of the attribute vectors.
> > >
> > >
> > >
> > > ## References
> > > [1] Qianli Ma, Jinlong Yang, Anurag Ranjan, Sergi Pujades, Gerard Pons-Moll, Siyu Tang, and Michael J. Black. Learning to dress 3d people in generative clothing. In *2020 IEEE/CVF Conference on Computer Vision and Pattern Recognition, CVPR 2020, Seattle, WA, USA, June 13-19, 2020*, 2020.
> > >
> > > [2] Qianli Ma, Shunsuke Saito, Jinlong Yang, Siyu Tang, and Michael J. Black. SCALE: modeling clothed humans with a surface codec of articulated local elements. In *IEEE Conference on Computer Vision and Pattern Recognition, CVPR 2021, virtual, June 19-25, 2021*, 2021.
> > >
> > > [3] Hugo Bertiche, Meysam Madadi, and Sergio Escalera. CLOTH3D: clothed 3d humans. In *Computer Vision - ECCV 2020 - 16th European Conference, Glasgow, UK, August 23-28, 2020, Proceedings, Part XX*, 2020
> > >
> > > [4] Hugo Bertiche, Meysam Madadi, Emilio Tylson, and Sergio Escalera. Deepsd: Automatic deep skinning and pose space deformation for 3d garment animation. In *2021 IEEE/CVF International Conference on Computer Vision, ICCV 2021, Montreal, QC, Canada, October 10-17, 2021*, 2021.
> > >
> > > [5] Yu Shen, Junbang Liang, and Ming C. Lin. Gan-based garment generation using sewing pattern images. In *Computer Vision- ECCV 2020- 16th European Conference, Glasgow, UK, August 23-28, 2020, Proceedings, Part XVIII*, 2020.
> > >
> > > [6] Alakh Aggarwal, Jikai Wang, Steven Hogue, Saifeng Ni, Madhukar Budagavi, and Xiaohu Guo. Layered-garment net: Generating multiple implicit garment layers from a single image. In *Computer Vision - ACCV 2022 - 16th Asian Conference on Computer Vision, Macau, China, December 4- December 8, 2022*, 2022.

---

> > > ### Author Response · Authors · 2022-12-06
> > > **Author Response to Reviewer 9oGC [1/2]**
> > >
> > > Thanks for your reply. We will revise our paper based on the rebuttal.
> > > We further discuss the concerns as follows.
> > >
> > > ## Q1: Dynamics can be learned from real-world data
> > >
> > > The key idea of our simulation-inspired method is to design a neural network to efficiently and effectively model the interactions between particles in a unified manner,
> > > thus dynamics in both synthetic data and realistic data can be learned by modeling the potential interactions based on the observation data without any prior.
> > >
> > > We train our model and learn the garments' dynamics on real-world dataset in the following.
> > > Specifically,
> > > we adopt the dataset generated from real-world scans mentioned in CAPE [1] and SCALE [2].
> > > The dataset in CAPE [1] and SCALE [2] only contains meshes for minimum clothed body and clothed body,
> > > where garments segmentations are not provided.
> > > To get the garment meshes,
> > > we remove body parts, which are not covered by garments, from the clothed body mesh, such as the head, neck, hands, and feet.
> > > We sample two types of outfits: blazer with long pants (BlazerLong) and long sleeves shirts with long pants (LongLong).
> > > The results are as follows.
> > >
> > > | L2 Errors (mm)  | BlazerLong      | LongLong        |
> > > | ----------------| ----------------| --------------- |
> > > | LayersNet       | $30.81\pm15.02$ | $38.63\pm22.65$ |
> > >
> > > Though the real-world dataset is noisy,
> > > our model still achieves similar Euclidian errors on the realistic dataset compared with synthetic datasets,
> > > such as Cloth3D [3] and our LAYERS.
> > > (Notice that in LAYERS,
> > > all objects are scaled up ten times the real-world size.)
> > > Our model achieves slightly higher errors due to the large areas of interpenetrations in the real-world dataset.
> > > We visualize some samples [**HERE**](https://youtu.be/Kdttw7LNiis).
> > >
> > > The main reason to adopt synthetic data
> > > is that they are easy-access with rich variations and high quality
> > > compared with real-world datasets.
> > > Generating realistic data with high quality is also more expensive
> > > than synthetic data.
> > > For example,
> > > in synthetic data,
> > > we can generate loose dresses and jackets with high quality,
> > > which are difficult to capture in the real world.
> > > More importantly,
> > > real-world datasets usually contain noisy data,
> > > such as large areas of interpenetrations between garments and human bodies,
> > > which can be alleviated in synthetic data.
> > >
> > > In addition,
> > > synthetic data are commonly used by researchers
> > > for tasks like garment animations [3,4], generations [1,2,5],  and reconstructions [6].
> > >
> > > Finally, the efficient data-driven methods are more suitable for many applications which need low latency and fast simulations,
> > > such as physics simulation in games or virtual reality.

---

### Author Response · Authors · 2022-11-25
**Looking Forward to Your Reply**

We thank the reviewers for your valuable time and insightful comments. We have tried to address all the concerns in the updated manuscript and our rebuttal text. We sincerely look forward to your reply. We are happy to answer any potential questions that are currently affecting your review. Thank you for your time!

---

Our main contributions are as follows:

* We propose LAYERS, a dynamic multi-layered 3D garment dataset, where different layers of garments are able to interact with each other constrained by physics laws and we introduce wind as extra driving factor besides the human body.
* We propose LayersNet and make the first attempt to animate garments by any kind of driving factors, which are human bodies and wind in this paper, from the view of simulation by data-driven method. Our method animates garments in a unified manner and achieves superior performance.

---

We briefly summarize the main updates in our revised paper

* We add and discuss some potential citations, including a recent dataset, in related work and Table 1. (Reviewer g4UX, X85t)
* We further compare with a dynamic garment animation method from "Motion Guided Deep Dynamic 3D Garments, Siggraph Asia 2022" in Section 5.2. Our LayersNet still achieves superior performance. (Reviewer 9oGC, X85t)
* We include more qualitative comparisons on LAYERS in Section A.2 in Appendix. (Reviewer 9oGC, X85t)
* We demonstrate the generalization ability of our LayersNet in Section A.3 in Appendix. (Reviewer X85t)
* We discuss the limitations of LayersNet in Section A.5 in Appendix. (Reviewer X85t)
* We upload a video as supplementary, including samples from our dataset and visualization results of our LayersNet. (Reviewer g4UX)

---

### Decision · Program_Chairs · 2023-01-20

**Decision:**

Reject

**Justification For Why Not Higher Score:**

Reviewers are not that excited about the work.  Both positive reviewers indicated they are okay with rejection.  In addition, the AC agrees the work has limited appeal to the ICLR audience and it would be better suited for a graphics / animation venue.  Finally, the AC does believe that the paper would benefit from a clearer indication that it is a combination of prior methods in the introduction and beginning of the methodology section.

**Justification For Why Not Lower Score:**

The work is well-executed and can potentially stimulate additional work in the area of multi-layer garment animation. The combination of the prior techniques is a novel combination and is also shown to be effective, and demonstrate new capability that was not previously possible.

**Metareview: Summary, Strengths And Weaknesses:**

Summary: The paper proposes a method to animate 3D garments by modelling interactions of cloth-body, cloth layers, and cloth-wind.  The proposed method combines data-driven particle-bases simulation (Shao et al), with patch-based garment representation (Ma et al), and uses a neural network to learn to decode the simulated patch embeddings associated with each vertex to obtain updated vertex position and velocity for the next time step.  A new large-scale synthetic 3D garment animation dataset (LAYERS) is also provided to study animation of multi-layer garments.

Strengths:
- The proposed method involves combining existing techniques to solve limitations of prior work in a practical and efficient way
- The proposed approach demonstrates capability not previously possible (modelling of multi-layer garments, effect of wind on cloth) and can stimulate additional work on data-driving animation of garments
- The proposed method shows clear superior performance against prior approaches
- A new dataset with realistic simulation is provided
- The paper is clearly written and well presented

Weaknesses:
- The writing (esp. introduction and beginning of section 4) should be improved to make it clearer that the proposed method is a combination of prior work
- Limited comparisons and qualitative examples
- Reviewers found the technical contribution to be limited as it is a combination of existing techniques
- The problem of multi-layered 3D garment animation is fairly specialized and may not be of interest to the broader ICLR community

**Summary Of Ac-Reviewer Meeting:**

It was difficult to find a time when all reviewers could meet.  The meeting was only with the more negative reviewer (g4UX), who expressed that their main concern was the limited technical novelty of the work, as it was a combination of existing techniques.  The revewer is not opposed to acceptance and agrees that the work can stimulate further research in the area of garment animation.

The positive reviewers (X85t, 9oGC) engaged in discussion on OpenReview, but could not make the proposed meeting time.  They indicated that they felt that the paper was well-executed and thus leaning positive.  However, both indicated they were okay with reject as well.